# Community Structures and Antifungal Activity of Root-Associated Endophytic Actinobacteria of Healthy and Diseased Soybean

**DOI:** 10.3390/microorganisms7080243

**Published:** 2019-08-07

**Authors:** Chongxi Liu, Xiaoxin Zhuang, Zhiyin Yu, Zhiyan Wang, Yongjiang Wang, Xiaowei Guo, Wensheng Xiang, Shengxiong Huang

**Affiliations:** 1Heilongjiang Provincial Key Laboratory of Agricultural Microbiology, Northeast Agricultural University, Harbin 150030, China; 2State Key Laboratory of Phytochemistry and Plant Resources in West China, Kunming Institute of Botany, Chinese Academy of Sciences, Kunming 650201, China; 3State Key Laboratory for Biology of Plant Diseases and Insect Pests, Institute of Plant Protection, Chinese Academy of Agricultural Sciences, Beijing 100193, China

**Keywords:** *Sclerotinia sclerotiorum* (Lib.) de Bary, diseased soybean root, antifungal activity, actinobacterial community, new agroactive compounds

## Abstract

The present study was conducted to examine the influence of a pathogen *Sclerotinia sclerotiorum* (Lib.) de Bary on the actinobacterial community associated with the soybean roots. A total of 70 endophytic actinobacteria were isolated from the surface-sterilized roots of either healthy or diseased soybeans, and they were distributed under 14 genera. Some rare genera, including *Rhodococcus*, *Kribbella*, *Glycomyces*, *Saccharothrix*, *Streptosporangium* and *Cellulosimicrobium*, were endemic to the diseased samples, and the actinobacterial community was more diverse in the diseased samples compared with that in the heathy samples. Culture-independent analysis of root-associated actinobacterial community using the high-throughput sequencing approach also showed similar results. Four *Streptomyces* strains that were significantly abundant in the diseased samples exhibited strong antagonistic activity with the inhibition percentage of 54.1–87.6%. A bioactivity-guided approach was then employed to isolate and determine the chemical identity of antifungal constituents derived from the four strains. One new maremycin analogue, together with eight known compounds, were detected. All compounds showed significantly antifungal activity against *S. sclerotiorum* with the 50% inhibition (EC_50_) values of 49.14–0.21 mg/L. The higher actinobacterial diversity and more antifungal strains associated with roots of diseased plants indicate a possible role of the root-associated actinobacteria in natural defense against phytopathogens. Furthermore, these results also suggest that the root of diseased plant may be a potential reservoir of actinobacteria producing new agroactive compounds.

## 1. Introduction

Sclerotinia stem rot (SSR) caused by a fungus *Sclerotinia sclerotiorum* (Lib.) de Bary is a highly destructive disease leading to serious economic losses to crops throughout the world. This fungus can infect over 400 plant species, including many economically important crops and vegetables [1,2,3]. Generally, the development of resistant cultivars is a long-term approach for controlling the disease [2,4]. However, the disease has yet been difficult to control because of the limited resource of the resistant genes. Therefore, fungicides have been used as the auxiliary method for controlling SSR in practice [5]. The benzimidazole and dicarboximide fungicides were the most efficient fungicides in controlling SSR [6]. However, the continuous use of these fungicides with high concentration can amplify the resistant level of phytopathogens [7,8,9]. Thus, development of new antifungal agents would be a constant need for controlling the disease.

Endophytic microorganisms residing inside plants have been found in majority of plant species [10]. A growing body of literature recognizes that some of these microorganisms are involved in plant defense against the phytopathogens through a range of mechanisms, including competition for an ecological niche or a substrate, secretion of antibiotics and lytic enzymes, and induction of systemic resistance (ISR) [11,12]. Recent studies on plant-microbe interactions reveal that plants can specifically attract bacteria for their ecological and evolutionary benefit by secreting root exudates [13,14,15]. It has even been postulated that plants can recruit beneficial microorganisms from soil to counteract pathogen assault [16,17]. For example, it has previously been observed that colonization of the roots of *Arabidopsis* by beneficial rhizobacteria *Bacillus subtilis* FB17 was greatly stimulated when leaves were infected by *Pseudomonas syringae* pv. *tomato* [18].

The phylum *Actinobacteria* consists of a wide range of Gram-positive bacteria with high guanine-plus-cytosine (G + C) content. Actinobacterial species are known to produce a vast diversity of active natural products including antibiotics, antitumor gents, enzymes and immunosuppressive agents, which have been widely used in pharmaceutical, agricultural and other industries [19,20]. Recently, endophytic actinobacteria have attracted significant interest for their capacity to produce abundant bioactive metabolites, which may contribute to their host plants by promoting growth and health [21,22]. A vast majority of endophytic actinobacteria have been isolated from a variety of plants including various crop plants, medicinal plants, and different woody tree species [23,24,25,26,27,28]. Further, recent cultivation-independent analysis using 16S rRNA gene-based methods revealed that actinobacteria can be specifically enriched in plant roots, and are more abundant in diseased plants than in healthy plants, which may provide probiotic functions for the host plants [29,30,31]. Thus, it is hypothesized that endophytic actinobacteria from disease plants may be a promising source for the discovery of new antifungal agents against *S. sclerotiorum*.

This prospective study was designed to test the above hypothesis by (i) using culture-independent and dependent methods to compare the generic diversity and antifungal activity of root-associated endophytic actinobacteria of field-growing healthy and diseased plants soybean plants and (ii) identifying antifungal metabolited produced by the outstanding actinobacteria isolated.

## 2. Materials and Methods

### 2.1. Plant Materials

Root samples were collected from soybean (cultivar: Hefeng-50) plants identified as SSR symptomatic (diseased) or asymptomatic (healthy) based on typical symptoms in a heavily infected soybean field in Suihua, Heilongjiang province, North China (46.63° N 126.98° E). The diseased samples showed lesions encircling up to 1/3 of stem diameter referring to a severity scale of three [32]. Each healthy plant and diseased plant located as close neighbors were determined as one group. The samples were brought to the lab in a cooler with ice in July 2017 and were processed immediately.

### 2.2. Isolation of Endophytic Actinobacteria

Three groups of root samples were used for isolation of endophytic actinobacteria. The root samples were air dried for 24 h at room temperature and then washed in water with an ultrasonic step (160 W, 15 min) (KH-160TDV, Hechuang, China) to remove the surface soils and adherent epiphytes completely. After drying, the sample was cut into pieces of 5–10 mm in length and then subjected to a seven-step surface sterilization procedure: A 60-sec wash in sterile tap water containing cycloheximide (100 mg/L) and nalidixic acid (20 mg/L), followed by a wash in sterile water, a 5 min wash in 5% (v/v) NaOCl, a 10 min wash in 2.5% (w/v) Na_2_S_2_O_3_, a 5 min wash in 75% (v/v) ethanol, a wash in sterile water and a final rinse in 10% (w/v) NaHCO_3_ for 10 min. After being thoroughly dried under sterile conditions, the surface-sterilized samples were subjected to continuous drying at 100 °C for 15 min. The sample was then cut up in a commercial blender and ground with a mortar and pestle, employing 1 mL of 0.5 M potassium phosphate buffer (pH 7.0) per 100 mg tissue. Tissue particles were allowed to settle down at 4 °C for 20–30 min, and an aliquot of 200 µL supernatants were spread on a series of isolation media and incubated at 28 °C for 2–3 weeks. Each isolation medium was supplemented with nalidixic acid (20 mg/L) and cycloheximide (50 mg/L) to inhibit the growth of Gram-negative bacteria and fungi. Five isolation media: Humic acid-vitamin (HV) agar [33], Gause’s synthetic agar no. 1 [34], dulcitol-proline agar (DPA) [35], cellulose-proline agar [36], and amino acid agar (serine 0.05%, threonine 0.05%, alanine 0.05%, arginine 0.05%, agar powder 2%, pH 7.2–7.4) were selected for the isolation. After 14 days of aerobic incubation at 28 °C, the actinobacterial colonies were transferred onto oatmeal agar (International *Streptomyces* Project medium 3, ISP3) [37] and repeatedly re-cultured until pure cultures were obtained, and maintained as glycerol suspensions (20%, v/v) at −80 °C.

### 2.3. Phenotypic and Molecular Characterization of Actinobacterial Isolates

The purified colonies were cultivated on ISP 3 at 28 °C for two weeks, and then grouped according to their phenotypic characteristics, including the characteristics of colonies on plates, color of aerial and substrate mycelium, spore mass color, spore chain morphology, and production of diffusible pigment. Those colonies with the same characteristics were classified as one species. The number of species was counted to compare the diversity of root-associated endophytic actinobacteria from healthy and diseased soybean.

Different phenotypic isolates were further subjected to 16S rRNA gene sequence analysis for the genus and species identification. The total DNA was extracted using the lysozyme-sodium dodecyl sulfate-phenol/chloroform method [38]. The primers and procedure for PCR amplification were carried out as described by Kim et al. [39]. The PCR products were purified and ligated into the vector pMD19-T (Takara Biomedical Technology, Beijing, China) and sequenced by an Applied Biosystems DNA sequencer (model 3730XL). The almost full-length 16S rRNA gene sequences (~1500 bp) were obtained and aligned with multiple sequences obtained from the GenBank/EMBL/DDBJ databases using CLUSTAL X 1.83 software. Phylogenetic tree was constructed with neighbor-joining method [40] using Molecular Evolutionary Genetics Analysis (MEGA) software version 7.0 [41]. The bootstrap method with 1000 repetitions was using to assess the topology of the phylogenetic tree [42]. Phylogenetic distances were calculated according to the Kimura two-parameter model [43]. The 16S rRNA gene sequence similarities were determined using the EzBiocloud server [44]. The obtained gene sequences have been deposited in the GenBank database.

### 2.4. Screening for Antagonistic Actinobacteria

The phytopathogenic *S. sclerotiorum* strain used in this study was kindly provided by the Soybean Research Institute of Northeast Agricultural University (Harbin, China). Antagonistic activity of isolates were evaluated through the dual culture plate assay [45]. The isolates were point-inoculated at the margin of potato dextrose agar (PDA) [46] plates and incubated for three days at 28 °C, after which a fresh mycelial PDA agar plug of the fungus was transferred to the opposite margin of the corresponding plate. After additional days of incubation at 20 °C for seven days, inhibition of hyphal growth of the fungus was scored. The inhibition rates were calculated as follows:Inhibition rate (%) = *Wi*/*W* × 100%(1)
where *Wi* is the width of inhibition and *W* is the width between the pathogen and actinobacteria. Each test was repeated three times and the average was calculated.

### 2.5. Isolation and Characterization of Antifungal Compounds

The antifungal compounds were isolated using an in vitro antifungal activity-guided method [47]. The active isolates were inoculated into 250 mL flask containing 50 mL of tryptone soy broth (TSB) [48] and cultivated for two days at 28 °C with shaking at 200 rpm. Then, 12.5 mL of the seed culture was transferred into 1 L Erlenmeyer flask containing 250 mL of the fermentation medium (soluble starch 1%, dextrose 2%, tryptone 0.5%, yeast extract 0.5%, NaCl 0.4%, K_2_HPO_4_ 3H_2_O 0.05%, MgSO_4_ 7H_2_O 0.05%, CaCO_3_ 0.5%, pH 7.2–7.4) and incubated at 28 °C for seven days with shaking at 200 rpm. The fermentation broth (20 L) was centrifuged (4000 rev/min, 20 min), and the supernatant and bacterial biomass were extracted with ethylacetate and methanol, respectively. Both extracts were concentrated by a rotary evaporator under reduced pressure until dry and mixed after dissolving their dried residues with methanol.

The crude extracts were divided into seven fractions using column fractionation packed with silica gel (200−300 mesh, Qingdao Marine Chemical Inc., Qingdao, China) eluting with petroleum ether/ethylacetate (20:1, 10:1, 5:1, 3:1, 2:1, 1:1 and 0:1). The bioactive fractions were then subjected to Sephadex LH-20 (Pharmacia, Uppsala, Sweden) and eluted with methanol to obtain several subfractions. The active subfractions were further separated by semipreparative HPLC (Hitachi-DAD, Tokyo, Japan) using a YMC-Triart C_18_ column (250 × 10 mm i.d., 5 µm) at a flow rate of 3.0 mL/min, and the potent active principles were finally isolated.

Structural determination of the active compounds were made according to spectroscopic analysis. NMR spectra were measured with a Bruker Avance III-600 spectrometer in CDCl_3_ or CD_3_OD using TMS as internal standard. The ESI-MS spectrum was taken on a Waters Xevo TQ-S ultrahigh pressure liquid chromatography triple quadrupole mass spectrometer. The HR-ESI-MS spectrum was acquired with an Agilent G6230 Q-TOF mass instrument. The UV spectrum was recorded in chloroform using a Shimadzu UV-2401PC UV-VIS spectrophotometer. The IR spectrum was obtained using a Bruker Tensor 27 FTIR. Optical rotation was determined in chloroform using a JASCO P-1020 polarimeter. The ECD spectrum was measured on a Chirascan circular dichroism spectrometer (Applied Photophysics Corporation Limited, Leatherhead, UK).

### 2.6. Antifungal Assay of Elucidated Bioactive Compounds

The active compounds were dissolved in methanol and diluted to different concentrations, which were then added to PDA medium. A fresh fungal plug of the fungus (5 mm in diameter) was placed in the center of the agar plate and incubated at 20 °C. Experiments were performed in triplicate, and the plate with the same amount of methanol was used as control. When the control plate was covered completely with fungal mass, the percentage of inhibition was calculated with the formula as follows:Inhibition (%) = (1 − *D*/*D_c_*) × 100(2)where *D* is average diameter of the treatment and *D_c_* is average diameter of the control. Data were subjected to linear regression analysis, and the effective concentrations required for 50% inhibition (EC_50_) were calculated.

### 2.7. Culture-Independent Community Analysis

Total community DNA was extracted from three groups of surface-sterilized root samples using FastDNA® SPIN for soil kit (MP Biomedicals, Solon, CA, USA) according to the manufacturers’ instructions. The purity and concentration of DNA were detected using NanoPhotometer spectrophotometer (Implen, München, Germany) and Qubit 2.0 Flurometer (Life Technologies, Carlsbad, CA, USA). Bacterial DNA pyrosequencing was based on ~460 bp amplicons generated by the PCR primers: 341F (5′-CCTACGGGNGGCWGCAG-3′) and 805R (5′-GACTACHVGGGTATCTAATCC-3′) with the barcode spanning the hypervariable regions V3-V4 of the 16S rRNA gene The PCR reaction was carried out in 30 µL reactions with 15 μL of Phusion High-Fidelity PCR Master Mix (New England Biolabs, Ipswich, MA, USA), 0.2 μM of forward and reverse primers, and about 10 ng template DNA. Thermal cycling conditions were 95 °C for 3 min, followed by 25 cycles of 95 °C for 30 s, 55 °C for 30 s, and 72 °C for 30 s, with a final extension at 72 °C for 10 min. PCR products was mixed in equidensity ratios. Then, mixture PCR products was purified with GeneJET Gel Extraction Kit (Thermo Scientific, Fermentas, Germany).

Sequencing libraries were generated using NEB Next Ultra DNA Library Prep Kit for Illumina (New England Biolabs, Ipswich, Massachusetts, USA) following the manufacturer’s recommendations and index codes were added. The library quality was assessed on the Qubit@ 2.0 Fluorometer and Agilent Bioanalyzer 2100 system (Agilent Technologies, Palo Alto, CA, USA). At last, the library was sequenced on an Illumina MiSeq platform and 250 bp paired-end reads.

When the sequencing was finished, we needed to filter the raw data to secure the quality, which mainly included the following steps: (1) Cut the polluted adapter, (2) remove low quality reads, specifically reads with average quality less than 19, based on the Phred algorithm, and (3) remove the reads with N base exceeding 5%.

According to overlap of the clean data, we spliced the paired reads by using the PEAR software [49] to merged sequences. The sequences were then removed Chimeras and clustered into operational taxonomic units (OTUs) by UCLUST [50] based on 97% pairwise identity. Taxonomic classification of the representative sequence for each OTU was done using a RDP classifier or QIIME’s closed reference strategy against the 16S rRNA gene database [51].

### 2.8. Statistical Analysis

Statistical analyses were performed via SPSS software, version 18.0. Means were compared via analysis of variance one-way ANOVA using the least significant differences test (LSD, *p* < 0.05). The data were reported as means ± standard deviation.

## 3. Results

### 3.1. Isolation and Distribution of Endophytic Actinobacteria

A total of 1574 endophytic actinobacterial colonies were successfully isolated. Based on their phenotypic characteristics, the colonies were preliminary classified into 70 species. Among the 70 species, 15 species were from healthy soybean roots, 27 species were from diseased soybean roots, and 28 species were shared by healthy and diseased soybean roots. The diversity of actinobacteria from diseased soybean root were greater than those from healthy soybean root. 16S rRNA gene sequence analysis of the 70 isolates revealed that they were distributed under 14 genera: *Streptomyces*, *Micromonospora*, *Actinomadura*, *Nonomuraea*, *Microbacterium*, *Rhodococcus*, *Promicromonospora*, *Microbispora*, *Kribbella*, *Mycolicibacterium*, *Glycomyces*, *Saccharothrix*, *Streptosporangium* and *Cellulosimicrobium* within the class *Actinobacteria*. *Streptomyces* was the most frequently isolated genus (58%, 41 isolates), followed by *Micromonospora* (5 isolates) and *Nonomuraea* (4 isolates). Some rare genera, including *Rhodococcus*, *Kribbella*, *Glycomyces*, *Saccharothrix*, *Streptosporangium* and *Cellulosimicrobium*, were endemic to the diseased samples (Figure 1). The 16S rRNA gene sequences were deposited in GenBank with accession numbers: MH919371–MH919374 and MN058215–MH058280.

### 3.2. In Vitro Antagonism of S. sclerotiorum and Identification of Bioactive Strains

Strains DAAG3-11, DGS1-1, DDPA2-14 and DGS3-15 exhibited strong antagonistic activity against *S. sclerotiorum*, with inhibition activity rates ranging from 54.1% to 87.6% (Figure 2). Strains DGS1-1, DDPA2-14 and DGS3-15 were from diseased soybean roots. For strain DAAG3-11, 176 colonies were from diseased soybean roots, whereas 11 colonies were from healthy soybean roots. Based on the 16S rRNA gene sequences, strains DAAG3-11, DGS1-1, DDPA2-14 and DGS3-15 were closely related to *Streptomyces sporoclivatus*, *Streptomyces cavourensis*, *Streptomyces capitiformicae* and *Streptomyces pratensis*, respectively (Table 1).

### 3.3. Identification and Activity Evaluation of the Antifungal Compounds

An antifungal activity-guided separation of the components of four active strains against *S. sclerotiorum*, using the in vitro antifungal assay, led to the isolation of nine compounds as their active principles (Figure 3). Out of the nine compounds, compounds **1** and **2** were from strain DAAG3-11, compound **3** was from strain DGS1-1, compounds **4**–**7** were from strain DGS3-15, and compounds **8** and **9** were from strain DDPA2-14. Compounds **1**–**8** are known compounds, which structures were elucidated as azalomycins F_4a_ (**1**) [52], azalomycins F_5a_ (**2**) [52], bafilomycin B_1_ (**3**) [53], actinolactomycin (**4**) [54], dimeric dinactin (**5**) [55], tetranactin (**6**) [56], dinactin (**7**) [56] and maremycin G (**8**) [57] by analysis of their spectroscopic data and comparison with literature values (Appendix A). Compound **9** is a new maremycin analogue.

Compound **9** was obtained as a yellow, amorphous powder. Its molecular formula C_22_H_27_N_3_O_4_S was determined by high resolution electrospray ionization mass spectrometry (HRESIMS) data (*m/z* 468.1555, [M + Na]^+^, calculated for 468.1564), corresponding to 11 degrees of unsaturation. The IR spectrum indicated the presence of hydroxy (3424 cm^−1^) and carbonyl (1720, 1682 cm^−1^) groups. The ^1^H NMR data (Table 2) of **9** suggested the presence of one 1, 2-disubstituted benzene system at *δ*_H_ 7.43 (1H, d, *J* = 7.8 Hz, H-4), 7.12 (1H, td, *J* = 7.6, 1.0 Hz, H-5), 7.37 (1H, td, *J* = 7.7, 1.2 Hz, H-6) and 6.89 (1H, d, *J* = 7.9 Hz, H-7). The ^1^H NMR data of **9** also revealed the presence of two methyl signals at *δ*_H_ 2.17, (3H, s, H-23) and *δ*_H_ 3.23, (3H, s, H-25). ^13^C NMR spectrum of **9** showed 11 sp^2^-carbons including three carbonyls at *δ*_C_ 204.84 (C-21), 178.33 (C-2), 168.5 (C-13) and eight aromatic or olefinic carbons at *δ*_C_ 152.27 (C-16), 142.83 (C-9), 130.36 (C-7), 130.01 (C-4), 125.53 (C-5), 123.24 (C-6), 109.08 (C-8), 100.43 (C-17). In the sp^3^-carbon region, the spectrum showed three methine at *δ*_C_ 42.84 (C-10), 52.94 (C-11), 52.8 (C-14), four methylene at *δ*_C_ 27.04 (C-18), 21.04 (C-19), 38.79(C-20), 38.75(C-22), and three methyl carbons at *δ*_C_ 16.36 (C-23), 8.92 (C-24), 26.65 (C-25).

Comparison the NMR data of **9** with FR900452 [58], an indole diketopiperazine motif linked with a cyclopentenone moiety, which was isolated from the fermentation broth of *Streptomyces* sp. B9173, implied that **9** was identified as a reduced form of FR900452 in which the cyclopentenone moiety is hydrogenated to cyclopentanone. As shown in Figure 4, the accurate assignments of all protons and carbons for compound **9** were preformed through the correlations in 2D-NMR spectra (^1^H–^1^H COSY, HSQC and HMBC, Appendix A). The HMBC couplings Me-25/C-9/C-2, H-5/C-3, and H-10/C-2/C-3/C-4, along with ^1^H–^1^H COSY correlations of H-5/H-6/H7/H-8, revealed N-methyl-2-oxindole unit. In addition, ^1^H–^1^H COSY correlations of H-18/H-19/H-20, as well as the HMBC cross peaks H-18/C-17/C-16, H-20/C-21, demonstrated that oxopiperazinyl moiety was linked to C-16/C-17 on the cyclopentenone moiety. ^1^H–^1^H COSY correlations of Me-24/H-10/H-11, together with the HMBC correlations from Me-24/C-3/C-10/C-11, indicated that N-methyl-2-oxindole unit was linked to C-10/C-5 on the oxopiperazinyl moiety. The ^1^H and ^13^C NMR spectroscopic data of **9** were also indicative of methyl mercaptomethylene moieties [*δ*_C_ 38.75, *δ*_H-CH2_ 3.17 (1H, m), 2.83 (1H, dd, 14.0, 8.2), *δ*_C-CH3_ 16.36, *δ*_H-S-CH3_ 2.15 (3H, s)]. The HMBC cross peaks Me-23/C22, H-22/C-14/C-13, along with ^1^H–^1^H COSY correlations of H-22/H-14, evidenced that methyl mercaptomethylene moieties were linked to C-14 on the oxopiperazinyl moieties, respectively. Therefore, the planar structure of **9** was elucidated as a reduced form of FR900452, depicted in Figure 3.

The relative configuration of compound **9** was determined by interpretation of its ROESY NMR spectrum. The correlations of H-14/Me-24, and H-10/H-11, indicated that H-14 and Me-24 were α-oriented, whereas H-10 and H-11 were β-oriented (Figure 4). Based on the close skeleton, the comparison of ECD spectra between **9** and N-demethylmaremycin B [57], and the largely consistent data supported that the configurations of 3-OH was α-oriented. Ultimately, the absolute configuration of was identified as 3S, 10R, 11R, 14R, resulting from the same trends of cotton effects (CEs) in the experimental ECD spectra of **9** and N-demethylmaremycin B.

The in vitro antifungal activity of compounds **1**–**9** against *S. sclerotiorum* was determined at various concentrations. All compounds showed significantly antifungal activity against *S. sclerotiorum* with the EC_50_ values ranging from 49.14 mg/L to 0.21 mg/L (Table 3). Thus, it further confirmed that these compounds were the main antifungal constituents produced by the four active strains.

### 3.4. Culture-Independent Communities

A total of 4116 OTUs containing 745708 high-quality reads were detected in the soybean root microbiome. The raw sequencing reads for this project were submitted to the National Center for Biotechnology Information Short Read Archive under accession numbers SRR8056376–SRR8056381. The predominant bacterial phyla were *Proteobacteria*, *Bacteroidetes* and *Actinobacteria* in the soybean roots. To compare the microbial communities obtained in healthy and diseased root samples from each group, the relative abundance of order *Rhizobiales* that can improve rhizobial nodulation and nitrogen fixation was significantly greater in the healthy samples, whereas the order *Actinobacteria* were more abundant in the diseased samples (Figure 5).

## 4. Discussion

The multifaceted approach adopted in this study, linking culture-independent and culture-dependent analysis, showed that actinobacteria were more abundant or diverse in the diseased soybean roots. This finding was in agreement with the previous study that the phylum *Actinobacteria* was higher in ‘*Candidatus* Liberibacter asiaticus’-infected citrus samples compared with that in healthy samples [31]. Another similar study also showed that potato plants infected with *Erwinia carotovora* subsp. *atroseptica* increased bacterial diversity [59]. The higher diversity of endophytic actinobacteria in diseased but healthy plants suggests that they may be involved in pathogen defense [60]. Indeed, extensive research has shown that endophytic actinobacteria has the capacity to control plant pathogens [22]. The in vitro antagonism assays demonstrated that four strains showed strong antifungal activity against *S. sclerotiorum*. Among the four antagonistic strains, all colonies of three strains were absolutely from diseased soybean roots, and another strain was also significantly enriched in diseased soybean roots compared to healthy soybean roots. A similar study has also shown that the rhizosphere soil of diseased tomato plant harbored a high percentage of antagonists [61]. Studies over the past few years have provided important information that plants possess a sophisticated defense mechanism by actively recruiting root-associated microbes from soil upon pathogen attack [18,62]. By adjusting the quantity and composition of its root secretion, plants can determine the composition of the root microbiome by affecting microbial diversity, density, and activity [63,64]. Our results seem to be consistent with previous observations. However, those strains with antagonistic activity in vitro may not be simply translated into biocontrol bacteria. Their biocontrol effects are influenced by various factors. For example, the antagonistic strains should reach a certain amount inside the plants to demonstrate a significant biocontrol effect [65,66]. Moreover, their secondary metabolite producing ability inside the plants may be influenced by plant physiological environment. Therefore, further work is required to assess the biocontrol efficiency in vivo and root-colonizing capacity of antagonistic strains by pathogen infection.

To learn more about the chemical nature of the antifungal activity, nine active compounds including six macrolides, two diketopiperazines and one 2-oxonanonoids, were finally obtained. Out of which, bafilomycin B_1_ (**3**) showed strongest inhibitory activity against *S. sclerotiorum*. Bafilomycin B_1_ has been reported to be produced by several *Streptomyces* strains and to show inhibitory activity against various fungi in vitro, such as *Rhizoctonia solani*, *Aspergillus fumigatus*, *Botrytis cinerea*, *Penicillium roqueforti*, and so on [67,68]. The antifungal activity of this compound against *S. sclerotiorum* was first reported in this paper. Azalomycins F_4a_ (**1**) and F_5a_ (**2**) were first isolated from the broth of *Streptomyces hygroscopicus* var. *azalomyceticus* [69]. Azalomycins F complex, including azalomycins F_3a_, F_4a_ and F_5a_ showed remarkable antifungal activity against asparagus (*Asparagus officinalis*) pathogens *Fusarium moliniforme* and *Fusarium oxysporumas* as well as powdery mildew pathogen *Botrytis* spp. [70]. The antifungal activity of the pure compound was first demonstrated in our research. Azalomycins possess broad-spectrum antibacterial and antifungal activities, and almost all of them were produced by *Streptomyces*, which were isolated repeatedly from soil and plant roots in the field by our laboratory (data no shown). This emphasizes the possible importance of *Streptomyces* producing azalomycins to protect plants against phytopathogens. A mixture of dinactin (**7**), trinactin and the major component tetranactin (**6**) is known as commercial pesticides polynactin (liuyangmycin in China), which can effectively control spider mites under wet conditions [71]. In addition, tetranactin (**6**) also exhibited significant antifungal activity against plant pathogen *Botrytis cinerea* with a minimum inhibitory concentration (MIC) of 24 µg·mL^–1^ [72]. Besides dinactin (**7**) and tetranactin (**6**), the monomer (**4**) and dimer (**5**) of polynactin were also isolated in this study, all of which were active against *S. sclerotiorum*. The antifungal activities of actinolactomycin (**4**), dimeric dinactin (**5**) and dinactin (**7**) have not been reported as yet. The findings reported here shed new light on the application of polynactin. Natural indole diketopiperazines exhibited a wide range of biological activities including antitumor [73], antibacterial [74], antifungal [75] and antiviral activities [76]. FR900452 is sulfur-containing indole diketopiperazines that showed specific and potent inhibitory activity against the platelet aggregation induced by platelet-activating factor [77]. Maremycin G (**8**) and compound **9**, structurally related to FR900452, showed significant antifungal activity against *S. sclerotiorum*. To our knowledge, this is the first report of the antifungal property of maremycins. Further research is needed to confirm the efficacy of in vivo disease control provided by the nine active compounds under laboratory and field conditions.

## 5. Conclusions

In summary, we report that soybean infected by *S. sclerotiorum* (Lib.) de Bary had a higher populations of actinobacteria and enhanced root colonization of antagonistic populations. In addition, eight known compounds and one new compound that exerted significant antifungal activity against *S. sclerotiorum* were obtained. These findings suggest that diseased plant samples could be a potential source for screening novel agroactive compounds, which contribute to a better understanding of plant–microbe interactions and provide new strategies for the development of agricultural antibiotics.

## Figures and Tables

**Figure 1 microorganisms-07-00243-f001:**
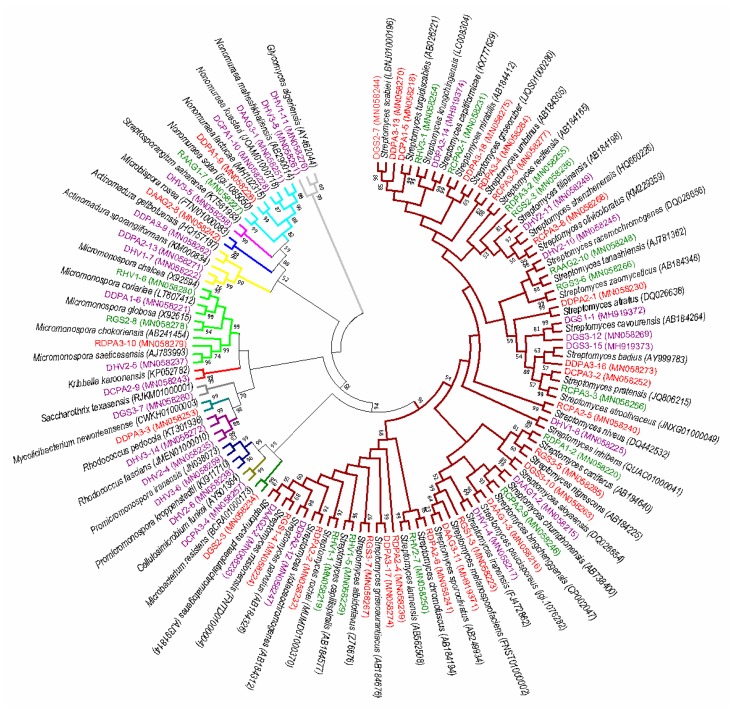
Neighbor-joining phylogenetic tree of 16S rRNA gene sequences from 70 endophytic actinobacteria in this study and their phylogenetic neighbors. Numbers at nodes are bootstrap values (percentages of 1000 replications); only values > 50% are shown. GenBank accession numbers of 16S rRNA gene sequences are shown next to isolate names. A branch indicated by the same color belongs to the same genus. Isolates indicated by green and purple are endemic to the healthy and diseased samples, respectively. Isolates shared by healthy and diseased samples are indicated with red.

**Figure 2 microorganisms-07-00243-f002:**
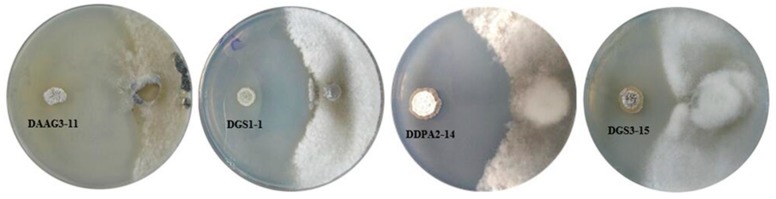
Dual culture plate assay between four endophytic actinobacteria against *S. sclerotiorum*.

**Figure 3 microorganisms-07-00243-f003:**
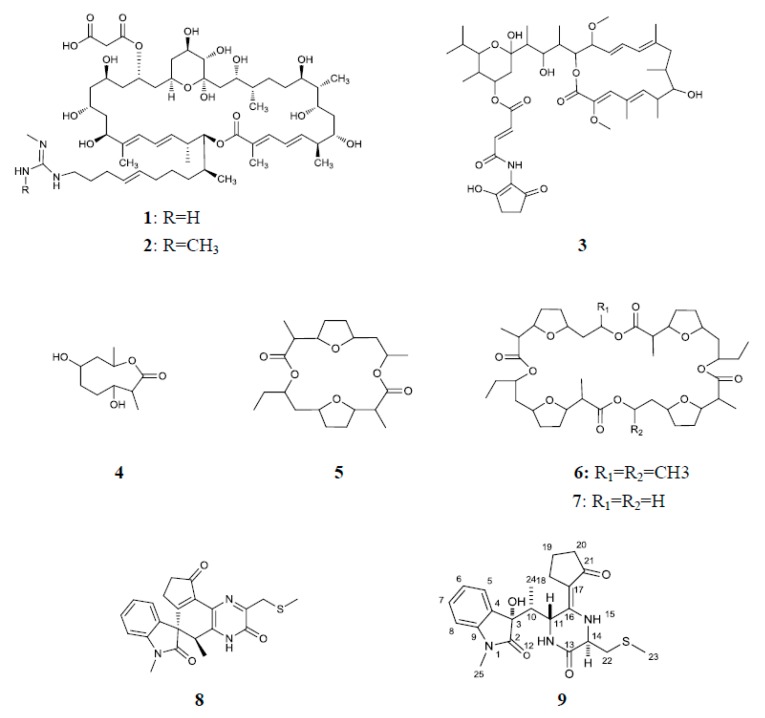
The structures of compounds **1**–**9**.

**Figure 4 microorganisms-07-00243-f004:**
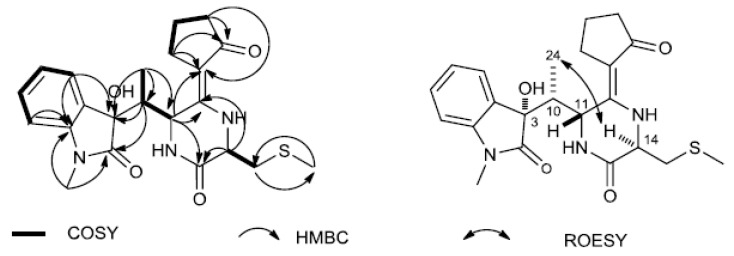
Key 1H–1H COSY, HMBC and ROESY correlations of compound **9**.

**Figure 5 microorganisms-07-00243-f005:**
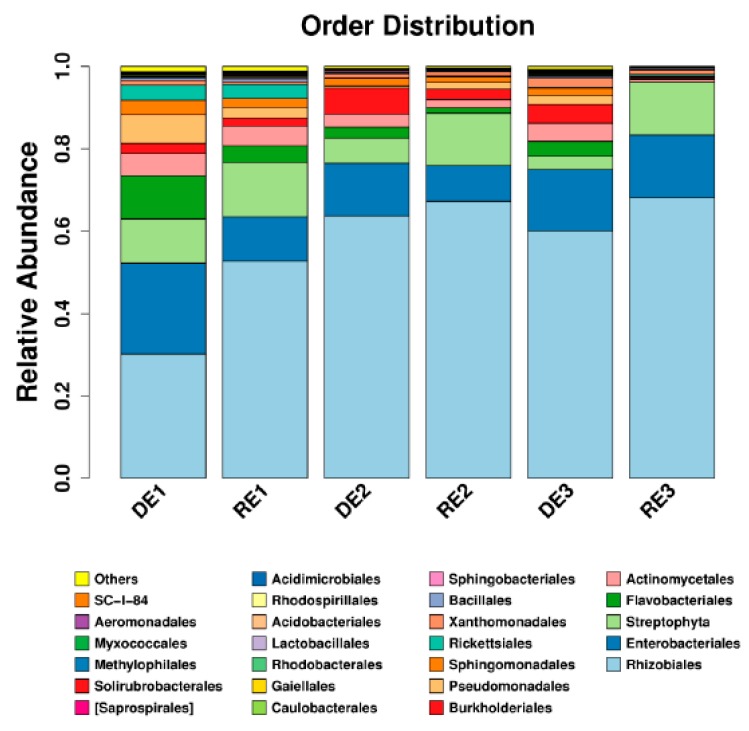
Analysis of culture-independent endophytic communities at order level in the soybean roots. DE, diseased sample; RE, healthy sample.

**Table 1 microorganisms-07-00243-t001:** Antagonistic potential endophytic actinobacteria isolated from healthy and diseased soybean root, and similarity values for 16S rRNA gene sequences.

Isolate No. and NCBI Genbank Accesion No.	Closest Type Strain with Accession Number	Similarity	Isolated From	Colony Number	*S. sclerotiorum* Mycelial Growth Inhibition (%) *
DAAG3-11 (MH919371)	*Streptomyces sporoclivatus* (AB249934)	100%	Healthy soybean root	11	87.6 ± 1.8 a
			Diseased soybean root	176	
DGS1-1 (MH919372)	*Streptomyces cavourensis* (AB184264)	99.9%	Diseased soybean root	13	78.9 ± 1.9 b
DDPA2-14 (MH919374)	*Streptomyces capitiformicae* (KX777629)	100%	Diseased soybean root	9	68.6 ± 3.4 c
DGS3-15 (MH919373)	*Streptomyces pratensis* (JQ806215)	99.9%	Diseased soybean root	6	54.1 ± 2.2 d

* Values are the means ± SE (*n* = 4). Data within the same column followed by different letters are significantly different.

**Table 2 microorganisms-07-00243-t002:** ^1^H NMR and ^13^C NMR data of compound **9** in CDCl_3_.

Position	*δ* _C_	*δ*_H_ (*J* in Hz)
2	178.33	
3	78.69	
4	130.01	
5	125.53	7.43 (1H, d, 7.8)
6	123.24	7.12 (1H, td, 7.6, 1.0)
7	130.36	7.37 (1H, td, 7.7, 1.2)
8	109.08	6.89 (1H, d, 7.9)
9	142.83	
10	42.84	2.06 (1H, m)
11	52.94	5.36 (1H, s)
12		11.01 (brs)
13	168.23	
14	52.8	4.14 (1H, dd, 8.2, 3.3)
15		7.09 (brs)
16	152.27	
17	100.43	
18	27.04	2.24 (1H, s)
		2.44 (1H, ddd, 13.9, 8.4, 3.8)
19	21.04	1.83 (2H, m)
20	38.79	2.28 (2H, m)
21	204.84	
22	38.75	3.17 (1H, m)
22		2.83 (1H, dd, 14.0, 8.2)
23	16.36	2.15 (3H, s)
24	8.92	1.19 (3H, d, 7.0)
25	26.65	3.23 (3H, s)

**Table 3 microorganisms-07-00243-t003:** EC_50_ values of active compounds against *S. sclerotiorum*.

Compounds	1	2	3	4	5	6	7	8	9
EC_50_ (mg/L)	4.87 ± 0.16 a	4.96 ± 0.13 a	0.21 ± 0.02 b	49.14 ± 0.82 c	5.33 ± 0.15 ae	3.69 ± 0.05 d	5.60 ± 0.11 e	3.46 ± 0.12 d	3.70 ± 0.05 d

Values are the means ± SE (*n* = 9). Data within the same column followed by different letters are significantly different.

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
