# Peer review of "Community Structures and Antifungal Activity of Root-Associated Endophytic Actinobacteria of Healthy and Diseased Soybean"

_microorganisms, 2019, doi:10.3390/microorganisms7080243_

Round 1

Reviewer 1 Report

The work presented is a well-done microbiological study balanced fundamentally and practically. The work is novel and original. It is devoted to the search of novel antifungal agents for protection of agricultural plants. Authors isolated and characterized more than 1,500 endophytic actinobacteria inhabited soybean roots. Only this fact provided with information about a diverse endophytic microflora saturated (14 genera) with actinobacteria. The dominance and abundance of actinobacteria in diseased plants including appearance of endemic genera was shown that could be an evidence of using antibiotic producers by plants as a defense mechanism against fungal pathogens. All actinobacterial isolates were tested for their antagonistic activity against Sclerotinia sclerotiorum, and the pure antifungal agents produced by these isolates were obtained. Among these agents, a novel compound with high antifungal activity was revealed.

The Introduction is written very clearly, and the study has a clear logic: isolation of bacteria from the targeted group – identification and characterization of isolates – estimation of their diversity – investigation of their biological roles (antifungicidal activity testing) in the ecosystem (roots of plants) – deciphering of mechanisms of their antifungal properties (extraction, purification, and structure analysis of metabolites) – confirmation of the biological role (testing of antifungal activities of purified metabolites) – estimation of the actinobacterial input in the whole microbial community. Moreover, various contemporary methods were used in this study including classical microbiological (enrichment on five isolation media, phenotypic analysis, and antagonistic tests on plates), molecular biology (sequence of 16S rRNA gene and metagenomics), and analytical (NMR, ESI-MS, IR and UV spectrum analysis, and optical rotation) assays.

Small remarks

The PDA (potato dextrose agar) abbreviation is deciphered (p4, line 150) after its first (p3, lines 115-116) appearance in the text.

It is better to give the names of solvents not as EtOAc and MeOH (p3, section 2.5) but as normal words “ethylacetate” and “methanol”.

It is not clear from the Materials and Methods from what parts of plants the actinobacterial isolates were obtained. In p2, section 2.2, only the reference on procedure is given. While in Summary (p1, line 19), there is a phrase “Endophytic actinobacteria were isolated from the surface-sterilized roots”. Apparently, “surface-sterilized” was important to underline the community of root tissues was studied. It is recommended to describe the procedure of strain isolation in more details.

A discussion question. Did you take into consideration that identification of isolates using data on the 16S rRNA gene sequences provided the identification of bacteria at the genus level only (Chun et al., Int J Syst Evol Microbiol 2018;68:461–466, DOI 10.1099/ijsem.0.002516)?

Summary. No doubts, the work is recommended for publication in the journal “Microorganisms” after very minor revision.

Author Response

Thank you very much for your constructive suggestions.
1. The PDA (potato dextrose agar) abbreviation is decipher ed (p4, line 150) after its first (p3,lines 115 116) appearance in the text.
R:We have revised (see lines 129 and 165).
2. It is better to give the names of solvents not as EtOAc and MeOH (p3, section 2.5) but as normal words “ethylacetate” and “methanol”.
R:We have revised (see lines 144, 146, 149 150, and l59 160).
3. It is not clear from the Materials and Methods from what parts of plants the actinobacterial isolates were obtained. In p2, section 2.2, only the reference on procedure is given. While in Summary (p1 , line 19), there is a phrase “Endophytic actinobacteria were isolated from the surface sterilized roots”. Apparently, “surface sterilized” was important to underline the community of root tissues was studied. It is recommended to describe the procedure of strain isolation in more details.
R:We have added the procedure of strain isolation in this revised manuscript (see lines 84 96).
4. A discussion question. Did you take into consideration that identification of isolates using data on the 16S rRNA gene sequences provided the identification of bacteria at the genus level only (Chun et al., Int J Syst Evol Microbiol 2018;68:461 466, DOI 10.1099/ijsem.0.002516)?
R:Our group have be engaged in antinobacterial polyphasic taxonomic identification for more than ten years. Thus, some isolates have been considered to be identified at the genus level as the next action plan. And we also think that the genome data for the taxonomy of prokaryotes is necessary.

Reviewer 2 Report

The manuscript by Liu et al. presents some interesting and unique results that would benefit the Microorganisms' readers. The paper is well prepared. Few of my comments and suggestions are addressed below.

Title:
Community Structures and Antifungal Activity of Root-associated Endophytic Actinobacteria of Healthy and Diseased Soybean

Abstract:
Lines 17-18: ... a pathogen, Sclerotinia ... roots.
Lines 19-20: A total of 70 endophytic actinobacteria was isolated from the surface-sterilized roots of either healthy or diseased soybean, and they distributed under 14 genera.
Lines 20-22: Some rare genera - can you give some examples of them? ... and the actinobacterial community was ... compared with that in ...
Lines 22-23: Culture-independent analysis of what and by which technique?
Lines 23-25: ... that were significantly abundant in ... - you can omit (Lib.) de Bary from now on - ... with the inhibition percentage of 54.1 - 87.6%.
Lines 26-27: ... the chemical identity of antifungal constituents derived from the four strains.
Line 28: ... were detected.
Line 29: Remove (Lib.) de Bary ... full definition (EC50) of 49.1 - 0.21 mg/L.
Line 30: ... more antifungal strains
Line 31: ... role of the ... against phytopathogens.

Introduction:
Lines 38-39: Remove all commas. ... caused by a fungus, Sclerotinia ... losses of crops ...
Lines 40-41: Generally, the development of resistant cultivars is a long-term approach
Line 42: ... has yet been ... limited resource of the resistant genes.
Line 44: Make this sentence present.
Lines 45-46: Resistance of what?
My suggestion: The continuous use of these fungicides with high concentration can amplify the resistant level of phytopathogens [6-8, please check the compatibility of these references].
Line 52: resistance of what, please clarify. ... interactions ... change 'have revealed' to 'reveal.'
Line 53: Remove own.
Line 54: It has even ...
Line 54: Please make tomato italic.
Line 58: Please provide the full definition of G+C.
Line 60: ... enzymes, ...
Line 64: Remove 'ecological.'
Line 68: ... for the host plants [28-30].
Line 69: ... from the diseased plants may be a promising source for the discovery of new ... - Remove (Lib.) de Bary.
Lines 71-74: ... to test the above hypothesis by (i) using both culture-dependent and independent methods to compare the generic diversity ... of root-associated endophytic actinobacteria of field-growing healthy ... soybean plants and (ii) identifying antifungal metabolites produced by the outstanding actinobacteria isolated.

Materials and Methods:
2.1 Plant Materials
Lines 77-79: It will be more understandable if the authors explain first how did they get the soybean plants from their field (indicating the field's geographical location). Be noted that North, not north. The following description can be how did the authors get the soybean roots.

Questions:
1) What do you mean by SSR?
2) Do you have the specimen voucher of your plant sample? Did you deposit your plant specimens to the acceptable national/international herbarium collection? Who did provide identification of your plant species, valid curator?

Line 81: I do not understand what do you mean by this sentence?

2.2 Isolation of Endophytic Actinobacteria
Questions:
1) Three groups, what are they? How were they different?
2) What do you mean by 'processed'? The authors need to explain how did they do. Surface sterilization? This method needs to explain briefly in the text, not just provide a reference. This question also covers what kind of sample that the authors spread on the isolation media. Remove 'for actinobacteria cultivation.' Gram, not gram. Move '(HV)' to be in front of agar. Add comma behind '[36].' Line 90 in parenthesis, agar powder.
3) What do you mean by 'were picked on oatmeal agar'? Please explain in details how did you select the actinobacterial colonies? How did you purify your isolates, and how did you preserve them for further studies?

2.3. Phenotypic and Molecular Characterization of Actinobacterial Isolates
Line 97: Add 'the' in front of same.

Questions:
1) Lines 97-99: I am wondering how the authors could classify your isolates based on their culture phenotypes into their species level. If I am right, this grouping is just probably at the generic level or streptomycetes and non-streptomycetes discrimination.
2) Lines 100-101: This statement is up to how complete of 16S rRNA gene sequences you analyzed. I don't think species identification is not necessary at this stage. Add 'the' in front of genus.
3) Lines 101-103: The word 'Chromosomal' sounds so specific. I think the authors referred to the total DNA as this term can include chromosome and plasmid. Remove '(2000).'
4) Lines 103-107: What about the size of the PCR products and in which region of the 16S rRNA gene? How did you align your sequences, CustralW or Muscle?

2.4 Screening for Antagonistic Actinobacteria
Lines 112-118: Remove '(Lib.) de Bary strain.' Provide the full definition of PDA. Add 'the' in front of fungus. Change 'fungal strain' to 'the fungus.'

2.5 Isolation and Characterization of Antifungal Compounds
Comment: Please check and use the same format of units; L, mL or l, ml throughout the manuscript.

Lines 124-125: Replace 'seed medium' with 'tryptic soy broth' and provide the reference of this medium. ... cultivated for 2 days at 28°C with shaking at 200 rpm.

Lines 125-128: Then, 12.5 mL of the seeded culture was transferred into 1-L Erlenmeyer flask containing 250 mL of the fermentation medium ... and incubated at 28 °C for 7 days with shaking at 200 rpm.

Lines 128-130: Please clarify the centrifugation unit, is it xg or rev/min, not rpm? Please change 'mycelia' to 'bacterial biomass.'

Lines 130-131: Both extracts were evaporated (how?) until dry and mixed after dissolving their dried residues with MeOH.

Lines 132-134: The crude extracts were divided into seven fractions using column fractionation packed with silica gel (200−300 mesh, Qingdao Marine Chemical Inc., China) and eluting with ...

2.6 Antifungal Assay of Elucidated Bioactive Compounds
Question:
1) MeOH and methanol, which form you prefer?
2) Potato dextrose agar (PDA) should be addressed at line 115, and the reference or source of this medium is required.

Lines 150-151: Please revise the sentence. What is 'the margin of medium area'?

Lines 152-153: You can use 'in triplicate.'

Lines 153-154: Please revise the sentence.

Line 154: The percentage of inhibition?

Lines 157-159: linear regression equations of what? Please revise the sentence.

2.7 Culture-Independent Community Analysis
Line 161: Total DNA ... What are those three groups?

Line 163: Please correct English usage.

Line 171: Please correct English usage.

Line 176: Remove '(Life Technologies, CA, USA)' as you already addressed in line 164.

Line 181: add and in front of (3).

Lines 182-183: PEAR software [47]. Finish this sentence at merged sequences.' and start a new sentence with 'The sequence ...'

Line 186: add gene in front of database.

2.8 Statistical Analysis
Please revise this paragraph. You compared means with ANOVA, or you compare treatments? What kind of ANOVA you used, one-way, two-way, ...?

Reviewer 3 Report

This manuscript is well written.

Author Response

Thank you for your support.